# Experiences of Opening Up Communicative Spaces for Large-Scope Issues

**Satu Kalliola \***  **and Tuula Heiskanen**

Faculty of Social Sciences, Tampere University, 33014 Tampere, Finland; Tuula.Heiskanen@tuni.fi
\* Correspondence: Satu-Sisko.Kalliola@tuni.fi

**Abstract:** The continuously changing world creates new challenges, large-scope issues, both at the community and the organizational level. Currently, sustainable development is among the key issues demanding organizational learning and new ways of operation. The paper looks for the potential of Scandinavian communicative-oriented action research (AR), applied in dialogue forums, to enhance learning and planning of integrative solutions to meet the needs of various actor groups. The paper links two intertwined AR lines of a Finnish work research institute to the contexts of classic and current AR discussion and their original social conditions in the early 1990s, when they were challenged by a severe recession. The characteristics of communicative spaces applied in the two cases are analysed qualitatively. The data, consisting of case reports, are reread and interpreted in a framework that concretizes Habermasian ideals of free communication. The elements of organisational learning and power embedded in the organisational positions of the participants dealing with large-scope societal issues are made explicit. Free communication and joint agreements of concrete plans require active agency that can be learned in a psychologically and socially safe communicative space where Habermasian lifeworld and system interact. The research shows the malleability of dialogue-based communicative spaces that can be applied in versatile social and organizational conditions. A future option would be a continuous dialogue applied in permanent dialogue structures.

**Keywords:** societal challenges as large-scope issues; action research; communicative spaces; organizational learning

## 1. Introduction

All societies face enormous challenges and need new ways to use all available resources—economic, material, and human —to meet them. These societal challenges, which are fundamentally large-scope issues, lead to constant changes at the community and organizational levels. Material production is strongly tied with ecological issues. Kasvio (2014) [1] sees that the wellbeing of societies and individuals is also tied to production that is economically, socially, and humanly sustainable, employs people, provides income, and takes care of their learning needs. Kasvio (2014) [1] continues that sustainable working life is one of the core factors of sustainability. This view calls for the definition of organizational sustainability used, for example, by de Waal, Weaver, Day and Heijden (2019) [2]: organizational sustainability results from activities that demonstrate the ability of the organization to maintain its business operations viably whilst not negatively impacting any social and ecological systems.

At the societal level Kemmis (2006) [3] argues for exploring education and community action for sustainability. At the organizational level de Waal, Weaver, Day and Heijden (2019) [2] put an emphasis on organizational learning and knowledge exchange. Specifically, they see the exchange of knowledge between the various parts of an organization as a main source of organizational learning, thus accentuating the importance of conscientious human resource development. These views bring forward the ideas of lifelong education

and learning as forms of sustainable human resource management as they relate to balancing human resources with organisational needs. This may be implemented formally by supporting staff to gain degrees, or, informally, through organisational learning that enables both the staff and the organisations they serve to keep up to date with changes.

As there is a current need for methods to offer organisational learning opportunities and to enhance knowledge exchange by crossing organizational borders, two Finnish action research (AR) cases, having addressed large-scope issues, are presented. The issues emerged during the severe recession of the early 1990s, in public sector work organisations. The cases have many connections and much relevance to the changing public sector of today, since needs always exceed economic resources, which are essential to the service production of the public sector in any modern society. The cases conducted among human resources specialists (Case 1) and day-care staff (Case 2) utilized the potential available in the organisations by bringing people together and encouraging them to learn how to make the changes seen as necessary for the new conditions. The AR approach chosen was a dialogical one, employing public spheres and democratic dialogue [3–9].

This article aims to shed light on how public spheres and democratic dialogue were combined in the cases to create communicative spaces where large-scope issues could be dealt with, within the boundaries of the type of management practiced. Also, the article aims to describe how these spaces contributed to learning as an important element in making changes.

Both cases started in the era of new public management (NPM) [10]. NPM was initiated to modernise the public sector, which was seen as excessively bureaucratic and producing problems in terms of job satisfaction and consequently in workforce availability. As the economic recession accentuated the costs of the public sector, there were retrenchments that left hardly any public organisations in Finnish society intact. Staff downsizing, accompanied by an NPM that emphasised managerialism, marketization, and privatisation, also challenged the traditional professionalism-oriented work culture of public service organisations. Among professionals there were wide discussions about how the government justified NPM by the necessity created by the severe recession, although the rationale may have been political or ideological. However, Pollitt and Bouckaert (2011) [11] (pp. 263–270) note that Finland did not have such strongly ideological governments with strong views about changing the role of the state as did the USA or the UK. Although the ultimate players [12]) were the government and the parliament, public sector officials at all administrative levels—from the central to the local—also had their say in the concrete downsizing of budgets. The cases started as options for customary budget cuts, accompanied by dismissals and layoffs. They were conducted at the Work Research Centre (WRC) of Tampere University, with Case 1 partly in cooperation with the University of Jyväskylä. The general aim, shared by the cases, was to learn to create new ways of working together with colleagues, managers, and researchers.

Olsen and Tikkanen only recently (2018) [13] found it relevant to refer to Fenwick's (2006) [14] notions of the dominance of individual learning and the lack of analysis of power in workplace learning studies, although work is most often a collaborative activity, indicating that this lack is still obvious. As a response to the challenges presented by Fenwick (2006) [14], we turn to Eikeland's (2012) [7] understanding of organisational learning in revisiting the cases. Eikeland (2012) [7] conceptualises the ways of learning as combinations of (1) individual or collective learning and of (2)theoretical learning or learning through experience. In this framework, the combination of collective learning and learning by experience is called organisational learning, which consists of what the members of the organisation do together, how they work in relation to each other, and how they work together to solve tasks. The types of work organisation and management practices may either enhance or limit the potential for learning, as the interrelationships present at workplaces include divisions of labour and power. We may summarise that organisational learning is dependent on the practical possibilities offered by organisational conditions. These tend to be tied to overall societal conditions that can also either limit or

enhance the potential for learning as work organisations interact with their surroundings, demanding and supporting change or maintaining traditions.

Eikeland (2012) [7] locates his notion of organisational learning in the communicative turn of action research (AR) conducted, for example, in Norway in the development of work organisations in the 1980s. The societal conditions for communication-oriented AR in large-scope working life issues were also favourable in Sweden, where organisational learning was pursued in dialogues involving labour market parties at the central and workplace level. The aim was to establish a permanent collaborative capacity to make changes in work organisations. In particular, the Swedish "Ledning, Organisation, Medbestämmande" programme (LOM or Leadership, Organisation, Co-determination) [8] was attractive to Finland, a latecomer to AR from an international perspective. The concept of democratic dialogue, applied in the LOM programme, aroused the interest of research institutions as well as labour market parties and the government due to the recession in the early 1990s. Since the recession was extremely severe in Finland, there aroused a widely accepted need for change as a matter of survival and the societal conditions encouraged learning and making organisational changes by dialogue processes [15,16]. Another line of theoretical influence came from researchers in the field of adult education who had applied AR in their educational activities [4,17], emphasizing critical approaches and emancipation as an objective.

Dialogical AR approaches, in giving voice to the people involved, have also attracted other researchers interested in large-scope issues. For example, Van Poeck and Vandenabeele (2014) [18] analyse how a multiplicity of views, values, interests, and knowledge claims can be taken into account in education in relation to sustainability issues. Rule (2004) [19] studies dialogue to fulfil emancipatory agendas in adult education research, whereas Rubenson and Elfert (2015) [20] express their concern about the adult education research agenda shifting away from classical themes of democracy and social rights. Recently, Wildemeersch and Fejes (2018) [21] pointed out the role of adult education in any matters of public concern. Against this background, this research on dialogical AR approaches in large-scope issues in work organisations is relevant. The article continues along the lines of Carr and Kemmis (1986) [4], Rule (2004) [19], Eikeland (2012) [7], and Gustavsen (1991) [8], who refer to Jürgen Habermas as one of their inspirations.

Although Habermas's (1984; 1987) [22,23] critical theory and ideas of communicative action are often seen as too abstract and idealistic [17]) to be applied in practice, ideality is also seen as a valuable starting point to create something new. We refer to Day (1993) [24] (p. 28), who sees that it is possible to use ideality, either in creative visualization or dreaming, to assist in emancipatory efforts, and that there are many techniques available in these efforts. For example, many action researchers appreciate Habermas's pursuits of free communication and the potentially resulting agreement, leading to action. On the other hand, action researchers have been criticized and are also concerned about the lack of theory and too much pragmatism when referring to Habermas [9].

In concretising Habermas's critical theory, Wicks and Reason (2009) [25] take as their focus how a collective of diverse individuals may effectively coordinate their action and orientations. They claim that Habermas's thoughts point toward a conclusion about the purpose of inquiry: it is to achieve agreement among human beings about what to do. They argue further that, in agreement with action research theory and practice, Habermas advocates for ongoing critical discourse among the members of a given community. Earlier, Frank (1989) [26], in his review of Habermas's ideas, also brought out the same aspects as Wicks and Reason (2009) [25]: the participants in communication should lay their conceptions of the "good life" and their validity claims open to critical reflection from others. In Habermasian [22,23] terms, this would mean that the participants are encouraged to bring their lifeworlds, referring to a person's subjective experiences and to what people experience together, into interaction with the system, referring to the strategic action of societies—for example, to formal organisations and institutional arrangements that are driven by money, power, efficiency, predictability, and control. Eikeland's (2012) [7] characterisation of com-

municative spaces, created along dialogical AR in work organisations, is along the same lines: participants can communicate freely about their accumulated work experiences, stepping out of the limitations of their systemically defined roles, and broaden the dialogue to an interaction situation, involving levelling the hierarchies of communication.

According to Habermas [22,23], lifeworlds and systems differ in the way they use communicative processes. Susen (2009) [27] argues that communicative processes may contribute either to the deliberative rationalisation of human coexistence and emancipation or to systemic rationalisation of human existence and domination. In other words, emancipation is related to communicative autonomy, when domination is intertwined with systemic steering and control.

With this as a vantage point, dialogical AR processes aim to ensure that different conceptions of the good life are mediated by conversation, producing shared experiences and mutual understanding, as Frank (1989) [26] has concretized Habermasian ideas. Frank (1989) [26] also encourages us to consider shared human experiences, and expressed validity claims, as lifeworld resources. They could provide for mutual understanding and consensus if the system does not cut off communicative interaction by orders. This approach, characterized with a basic conflict between lifeworld and system, appears as complex in the practical world. Lifeworlds may conform to the system or may consist of smaller systems—for example, managerial or professional ways to use communicative processes with citizens, clients, or representatives of other professions. All in all, when individuals contemplate whether to express their thoughts to others and especially to system representatives, they may find the situation uncomfortable and will not use any opportunities offered to express their views.

Gustavsen (1991) [8] has addressed the above-mentioned crucial aspect of free communication in the criteria of democratic dialogue. The criteria emphasise both the right and the obligation of all members of work organisations to give their input in the process and to accept that other participants may have better arguments than their own in the continuing, ethical pursuit of the generation of joint action. Although a critical approach and emancipation are not mentioned, they are implicit in the criteria that may be seen as one of the AR tools, mentioned by Dick (2015) [28] (p. 436), to increase the extent, and also the depth, of participation.

Frank (1989) [26] further states that Habermas argues for continuous communication since the process of exchanging validity claims always generates new social situations and so there can be no final solutions. Thus, at its best, dialogical AR continues among the participants after a formal research project is over. Most importantly, Frank (1989) [26] emphasises Habermas's arguments about the essential problem of politics as being communicative, and implies that communication has the potential to coincide diverse conceptions of better life.

We find the definition of a communicative space by Kemmis (2001) [5] to be in congruence with Habermas's basic ideas of free communication. Kemmis (2001) [5] (p. 100) sees a communicative space to be constituted when issues or problems are opened up for discussion and when participants experience their interaction as fostering democratic expressions of diverse views and as permitting the achievement of a mutual understanding of what to do. Although the concept of power is included in the thoughts of Habermas as part of a system, it is only implicit in Kemmis's definition, emphasising free communication. It is not possible to achieve free communication without tackling the issue of organisational power in some way or another. Also, learning is implicit in Kemmis's definition as a continuous exposition of diverse views that are an abundant source of learning for all participants. Since, in the context of practical development work in work organisations, power [14] and learning [7] cannot be ignored, we see it as useful to study communicative spaces in the framework of these three intertwined dimensions: free communication, power, and learning.

There are, nowadays, many contributions addressing the issues of organisational and societal development by means of AR variations [28,29] However, we see a need to gain a better understanding of the concrete details of the applications and their related outcomes.

The research task is to describe and analyse the cases in a new conceptual framework consisting of free communication and organisational learning, made available by communicative spaces and power embedded in the work organisations. The basic characteristics of the communicative spaces of the two AR cases, the practical outcomes with their prerequisites or constraints, and the learning outcomes for participants and action researchers are defined. The various characteristics of the cases are highlighted to offer an understanding of the conditions for the outcome. The article proceeds to a section describing the data and analysis, the following section presents the findings, and the article ends with a discussion and concluding remarks.

## 2. Methods, Data, and the Cases

In this study Rapoport's (1970) [30] classic notion of AR's dual agenda, benefiting from both practice and research, is acknowledged. This means taking AR as a research strategy that, in addition to allowing the researchers to work together with public sector work organisations in their search for new modes of operation, also allowed them to gather and analyse data in multiple ways. Often AR necessitates reporting on practical outcomes and how they were attained before further conceptual and methodological elaborations are made in scientific publications. Also, progress and final reports aim to document the practical outcome and the AR processes in detail, since they are important to participants due to their anticipation of improvements in the initial phase [28].

During the AR processes, learning-oriented AR circle models, with a focus on the evaluation by Lewin (1948) [31] and on a critical reflection by Carr and Kemmis (1986) [4], were applied. The baseline in the beginning, the progress, and the outcome were evaluated by case-specific questionnaires or quality of working life surveys, interviews, and document analyses. The researchers have followed the AR processes of the cases by participant observation documented in researchers' diaries and have been involved in continuous reflection or evaluation with the people involved.

Research ethics were followed. The participants were involved voluntarily, their anonymity was maintained, and, going along with the practice of AR, they were informed of the interim and final results in seminars before the results were published; thus, they were able to give feedback, when needed. Also, the participants were aware that the researchers observed the search sessions and made notes (kept diaries) to be able to understand the situation at hand, but no limits were given concerning the phenomena to be observed. Reciprocally, the researchers were obliged to follow confidentiality in participants' personal matters. If mutual trust could not be established from the start, potential organizations and individuals refused to participate in the project.

Following Rapoport (1970) [30] and Dick (2015) [28], the opportunity to use the material produced along with conducting and reporting the cases as data was taken. The qualitative data consist of researchers' accounts of past AR processes, final reports, journal articles, and other publications that have described, utilised, and reported original empirical data related to the research task, with the exception of some original interviews in Case 1. All publications used as data are listed in the references.

The research task calls for a descriptive methodology. The data are examined qualitatively in light of the chosen framework to find relevant themes for a detailed description of the characteristics of both cases [32]. Firstly, in the section regarding descriptive reading (Section 3.1.1), descriptions are compiled of what was actually undertaken in the cases: how action research was conducted when creating and applying communicative spaces. The same approach was used in the compilation of the descriptions of practical and learning outcomes (Sections 3.2 and 3.3). These results may be interpreted as action research narratives, guided by the interplay of provisions for free communication, redistribution of power, and learning in the cases. Secondly, in analytical reading, the key concepts of

the chosen framework—free communication, learning and power—are broken down to their essential characteristics and the focus is transferred to empirical phenomena and practices as reported in the material of the cases, as a flow from the abstract towards the concrete, and vice versa. The characteristics used are the structures of communicative spaces, including bases for participation; actualisation of the elements of power, including ways to reach mutual agreements; and expressions of learning. The data of both cases are examined parallel to give also a possibility to comparisons, when seen appropriate (Section 3.1.2). Methodologically, this article is a result of a qualitative second reading of the data in a new framework guiding the interpretations.

Due to ongoing changes affecting public sector work, Case 1 started by offering a further education programme to experts and professionals employed in the Finnish public sector [33]. As a new type of activity initiated by university researchers, the programme received substantial financial support from public resources and only minor costs were covered by the participants' organisations. The case project lasted in different forms throughout the 1990s [34]. The recruitment for the education programme started in the autumn of 1990, the first gathering took place in March 1991, and the final session was in September 1992 (duration: 18 months) after which the project was carried out as specific further research and personnel education work. The 28 participants were selected among applicants working in central, district, and regional administrations, various departments of the local government, separate expert organisations, and institutions of higher education engaged in management, planning, or training tasks, as well as in grassroots-level professional service work. The importance of the ongoing development project was stressed as a link between the programme and the work organisations. The research group was interested in contributing constructively to expert and professional work in the midst of the transformation of the whole public sector. The reasoning behind the focus was to bring together key groups whose work tasks included the interpretation and anticipation of changing situations. The research group, with earlier experience of studies in work organisations facing challenging changes, offered the programme to support such interpretation work and to make it easier for work organisations in the public sector to find a new perspective on the surrounding reality. It was presumed that participants would present the pressures for change that were affecting the public sector into the general discussion of the educational forum.

Case 2 started as a single project involving kitchen staff ($n = 150$) from day-care centres ($n = 98$) in a city in 1997–1999. The kitchen staff felt overwhelmed by continuous change that weakened the previous autonomy of the staff and introduced performance measurements and computers into kitchen work. A chief shop steward contacted the researchers as a liaison and outlined the initial theme of the project within the frameworks of quality of working life and productivity, both of which were deteriorating due to the fiscal crisis and the chosen means to survive it among day-care kitchen staff. The theme of computer skills continued to a second phase involving kindergarten aides ($n = 535$) in 2001–2003, while the theme of performance was the focus of a pilot project in a transition towards a contractor model, involving the staff of five (5) day-care centres ($n = 270$). Since then, the city of Case 2 has almost continuously taken part in Network A or its spin-offs, most recently in 2012–2015, and up to the present in a dialogue project funded by the European Social Fund (ESF) [35].

## 3. Results

### 3.1. Creating and Applying Communicative Spaces

3.1.1. Descriptive Reading: AR Approaches Tailored for the Aims and the Characteristics of the Cases

In Case 1 the pursuit of free communication [22,23] was carried out in compliance with AR approaches used in the field of education e.g., [4]. The researchers wanted to rely on the reflective potential of the participants in their involvement in the improvement of the practices of public sector work organisations from the very beginning. Before the start

of the programme, interviews were conducted to gauge the participants' expectations of the programme.

The preliminary assumption was that the participants as individual experts were responsible for whole operational sections and their interpretations of the direction and alternatives to the ongoing process of change would affect the future shape of the public sector. Since the backgrounds and occupational tasks of the participants varied greatly, the specific expectations were diverse. However, three kinds of expectations were quite common. Firstly, the participants wished to find new ways to carry out their tasks. Secondly, they wished to fulfil departmental expectations, and thirdly, they had learning-centred reasons and a desire to renew their knowledge of research on working life. The desire to network was also raised in a number of interviews.

"Knowledge becomes outdated rapidly. One has to learn new [things] . . . one should have the bigger picture [in development work]."

"I hope to receive a broader background for the development plan related to my work."

"My ambitions [for this programme] relate to my own work, the benefits I will receive for my work."

(Excerpts from three interviews conducted before the start of the programme.)

The content and methods of the programme were left partially open at the start. The reasoning behind this choice was to utilise the opportunity to modify the content and procedures according to the challenges and problems brought up in the forum by the participants based on their situations and realities.

The programme consisted of four main working methods during the 18-month period: communicative spaces consisted of large seminars and small group discussions that were complemented by independent self-regulated studies, and individual work in the participant's own job and organisation, where the lessons learned were applied (for more details, see [33,36]). The large seminars were events where all participants, researchers, educators, and tutors gathered to have dialogues on the challenging societal conditions and ways to respond. The design of the programme according to the participants' wishes was realised in the topics of the lectures that the educators provided for the forum. The topics dealt with management and development issues in the public sector as well as learning, expertise, and promises of multidisciplinarity, and they provided material for the specific themes to be worked on in small groups. The time schedules of the sessions ensured room for questions and debates and the organisers also encouraged open communication.

Both during the large seminars and between them, the participants worked in small groups supported by two or three tutors. These small groups were multiprofessional, consisting of people from different levels of the public sector. Each group had a different meta-level generative theme that was planned to support the framing of the participants' development projects and which functioned as a link between the forum and participants' organisations. In order to facilitate the building of a link between the programme and the organisations concerned for the first large seminar, a member of management—usually the immediate superior of the participant—was invited, in addition to the selected participants. For research purposes, all the discussions in the large seminars and small groups were tape-recorded and transcribed. The project reports compiled during the programme concerning organisation or job-specific development projects were shared with the participants and their supervisors and also used as research data, as were the participant interviews made at the start and end of the programme. These data were also utilised in later years.

Case 2 was hosted by Network A, which adopted democratic dialogue and its application in dialogue conferences [8] as the core AR method to organise communicative spaces. The criteria of democratic dialogue and the design of the conferences were presented to the potential participant organisations in preliminary negotiations in every case. The conferences, together with quality of working life surveys and case-specific measures, were offered as a method to be used in the framing of the projects' objectives, in promoting free communication in interim and final evaluations, and in planning steps to be taken after the project. The role of the researchers was as facilitators. The criteria of democratic dialogue

do not mention learning specifically, but it is inherent in the process and expressed as a demand for all participants to have the opportunity to gain an understanding of the topics under discussion [8] (pp. 290–291).

These criteria were delivered to the participants along with the conference invitation in order to stress their significance, although Gustavsen (1991) [8] (p. 294) stresses an idea of emerging democratic dialogue. The criteria were present in the conferences, sometimes as posters on the walls of the conference rooms and sometimes as leaflets circulated among the participants, and their characteristics were discussed at the beginning of the conferences. The application of democratic dialogue was also encouraged in representative task forces (all hierarchical levels; all professions; sometimes with partners in development crossing the organisational borders), in all workplace or department meetings, and in small groups working on special assignments. In cases where a steering or advisory group was also established, trade union representatives, members of the municipal boards, and other stakeholders were invited to become members [37] (pp. 38–40).

As the initiator of Case 2, a chief shop steward wanted support for kitchen staff experiencing a very high level of sickness absenteeism amidst continuous management cutbacks. Permission to start the project was given by a senior manager, who accepted the need to balance the relationship between the quality of working life and the performance requirements in the day-care kitchens. Interviews and quality of working life surveys in 1997 provided information about a very hierarchical management with hardly any staff involvement. The staff were heard only through their trade union and occupational safety representatives in the first phases of the project. It was evident that the kitchen staff did not have any opportunities to exert their agency [38] in any matters concerning their work and working conditions. The project started by establishing a formal representative project group that coordinated all the development issues. Also, representative task forces on the current challenges raised by the changes were formed. At this point, the management appointed a day-care supervisor as a liaison person with delegated authority to promote new modes of operations. There were 13 task forces concentrating on the new contents of the kitchen work, in-house control of food safety, the use of information technology in the planning of the menus and inventories, an increased number of allergic children with special diets, and amelioration of the workplace stress accentuated by the low number of staff due to sickness absenteeism. One task force focused on the issue of performance measurement, which was the most stressful because it was used to implement the new cutbacks. In addition, a day-care manager and a kindergarten teacher were appointed to plan concrete ways to improve cooperation between the kitchen, children, and other staff. Since the day-care organisation consisted of five regions, regional workplace meetings of 20–25 day-care centre kitchens were organised regularly on an occupational basis. All project bodies were encouraged to practice democratic dialogue to look for common ground in the changes. After these dialogue exercises leading to concrete change action, dialogue conferences were organised in the final phase of the project to evaluate the project results and to plan the next steps. These were regional and conducted as five separate conferences [39].

The aims of both cases are coherent with the essential aims of AR, as stated by Carr and Kemmis (1986) [4]: they have aimed to involve those concerned and to improve the practices by a new understanding of the practitioners about what they are doing in given conditions. New understanding emerges from dialogue among the participants in communicative spaces created along AR processes.

Furthermore, the two AR lines converge in their basic approaches to AR. Martin (2008) [12], agreeing with those who see learning—and a developed capacity to learn—as the core result of AR, differentiates two major challenges that large-scope AR projects face. One is making sense of the systems, determining what systems will experience an impact due to the change, and determining who the ultimate players are in the large-system change. The other is designing and implementing processes that engage multiple perspectives and support inquiry and learning (see also [7,25,40]). In summary, AR was

valued in both cases in accordance with Greenwood's (2015) [29] notion of it as a strategy that allows for the use of multiple theories and methods in gaining new knowledge and promoting change simultaneously.

3.1.2. Analytical Reading: Characteristics of the Cases along the Dimensions of the Conceptual Framework

Although the cases converge in many aspects of AR approaches, as found by a description, they have differences when analysed more closely in the chosen framework [32]. The analysis points to differences in all the dimensions of the chosen framework. The relevant themes in carrying out free communication [22,23] are how communicative spaces were structured, how participant recruitment was organised, and how the agenda to be discussed emerged. The dimension of organisational power [14] was analysed, asking how power was present in the dialogues, and the dimension of organisational learning [7] was approached through its explicit characteristics in communicative spaces.

- Structures of communicative spaces The basis for participation in Case 1 was that of voluntary professional and personal interest of public sector professionals, who needed permission from the employer or top management to take part in the programme. The programme offered them dialogues in large seminars (dialogue conferences), small group discussions, and interaction with independent self-regulated studies. Also, independent work was included. In Case 2, the participation of individuals was also voluntary. However, after an organisational decision to start the project along the lines of democratic dialogue, all the occupational and hierarchical groups, from staff to top management, clients, staff representatives, and political decision makers, were invited to assign the members to project groups/task forces, workplace meetings, occupational meetings, and regional dialogue conferences. In the framing of the agenda of communicative spaces, the cases had different sources of topics for dialogue when they searched for ways to cope with large-scope issues. In Case 1 they were chosen by the participants and researchers together as an evolving dialogue, whereas in Case 2 the main themes of the research and development plans were agreed in the preliminary negotiations. Task forces modified them according to the results of surveys, interviews, and dialogue conferences.
- Actualisation of the elements of power The analysis pointed out how power expressed itself in the authority possessed by the participants and in the organisational status of the communicative spaces during and after the projects. As the participants of Case 1 were individuals from separate organisations, the programme offered them a continuous opportunity to apply new thoughts, including the idea of transferring communicative spaces in their own organisations. However, a negotiation with top management was needed. In Case 2 all individuals came from one organization but possessed varying degrees of official authority. Also, communicative spaces had power delegated to them as a whole to carry out the agreements reached in dialogues. The communicative spaces aimed to gain a permanent position as a development forum.
- Organisational learning as learning from others and together When organisational learning is understood as consisting of what the members of the organisations do together, it becomes important to really hear others, which is one of the key elements of a dialogue, after the participants have exposed their convictions. In Case 1 this was complemented by individual reflection through independent studies, with the aim of learning to act as change agents in their organisations. In Case 2 the development process also included learning through evaluation, learning to integrate development and change into everyday work at individual and organisational levels, and relying on the potential of learning together as a working community.
- Learning to use agency as a prerequisite for organisational learning In reflecting on the concrete ways of organisational learning in the communicative spaces provided by the cases, there was one more major difference: agency [38]. The initial premise in Case 1 was that the participants would see it as their responsibility to take the

results gained from the programme back to their work organisations as they were already in positions that both allowed and demanded the use of their agency. As the participants and researchers worked on a voluntary and spontaneous basis, it was each individual's responsibility and choice to use their voice. In Case 2, the criteria of democratic dialogue as normative rules denied the role of expert authority to the researchers. Instead, they aimed to offer equal opportunities to express one's own perspectives to all and encouraged participants to use their voice. This presented tasks and challenges for individual learning for those participants who were used to working under supervision.

*3.2. The Practical Outcomes and Their Prerequisites and Constraints*

Case 1 started with a large seminar, a dialogue conference, where superiors from the participants' organisations were also present. This was supposed to ensure that the participants were empowered to integrate the programme into the development projects in their own work. In most cases, this wish came true, and a clear majority of the participants felt that their participation in the programme was valued in their workplace. However, there were also some who did not get such support. The project types that the participants worked on during the programme included (1) developing work communities, (2) evaluating a developmental operation, (3) identifying social obligations of the operational idea of an organisation, (4) examining roles and links between service organisations, and (5) identifying the need for and targeting services.

All but six participants compiled a project report on their work in the development project and its results. The clearly expressed constraint on completing the programme was a lack of support from supervisors, thus implying the relevance of power embedded in the hierarchical positions. Another constraint may have been the lack of individual motivation, but this was not assessed in the case. On the other hand, individual motivation and interest in their own work may have been motivating factors for a majority of participants, anticipating that their work in the programme would have effects on their own work and/or the organisation:

"[Integrating personal and organisational interests] has worked really well. This programme has not at all remained detached . . . what I have done in this programme has become directly useful in my work."

"Short pieces of the report we did on the programme have been included in quite a number of papers . . . it has found many users."

"A network was created . . . it is much easier to do one's work when one has a network."

(Excerpts from three interviews conducted after the programme.)

It may be concluded that the practical outcome for the participants in Case 1 was to learn to act as change agents in their organisations. The new matters that were reflected on and learned about were either acknowledged as directly applicable to their work or disseminated in organisational documents for a wider set of actors. Moreover, a new possibility for collegial support from the participants' network was appreciated.

In Case 2, in-house control of food safety turned out to be easy when all other issues were either processed or on their way to being processed. All the main users of the new technology received relevant education, more computers were provided for the kitchen staff, and the issue of food allergies was covered partly by dietary education and partly by centralisation. The kitchen staff learned to cope with their stress as they talked about it. The many communication spaces offered to them in small groups and regional meetings led them to express in words their daily, weekly, and monthly schedules, and their supervisors learned to listen to them. These phenomena may be interpreted as first steps towards leadership by dialogue [41], which in Habermasian terms is one way to use communication as a means of deliberating different views to promote emancipation [27]. In addition, members of the kitchen staff were empowered to ask for help when they needed it. The performance measurement task force outlined its planning process and gathered opinions

via email. It started with the technicalities of renewing the forms and the division of tasks in collecting data. Many compromises were made in the actual contents of the performance indicators and their coefficients but, in the end, a new plan was introduced with the next budget. In the third theme—cooperation between kitchen and other day-care centre staff— a good grounding was achieved in discussions about the various roles and tasks of the working community. After that, in the autumn of 1998, half of day-care centres organised their own theme week, for example, preparing the children's favourite meals or planning the whole weekly menu with the children. In the spring of 1999, almost all the day-care centres planned this type of activity through joint discussions and carried it out in an improved atmosphere [39].

The case achieved all the practical goals it had pursued. This contributed to the responses given in the quality of working life survey at the end of the project. For example, when compared with the first survey, a larger proportion of the staff was satisfied with the content of the kitchen work and their opportunities to use discretion and get feedback and support. About 90% of the respondents identified at least one positive change produced by the project, and no negative changes. The mean level of sickness absenteeism, in days per employee per year (1997–1998), decreased from 23.4 to 19.9 [39] (p. 70). Turning the words expressed in many dialogue forums into action seemed to be a prerequisite for the good results. It also continued in the next phases of Case 2, which expanded to include other activities of day-care centres in the city [42,43]. Along with the dialogues, the members of the staff had learned to express their views, while the top managers, for their part, learned to listen and to take into account the views of the staff, thus involving the staff in decision-making and power issues. This combination of individual learning on different hierarchical levels and in different power positions of the organisation led to organisational learning and was concretised in making the planned changes together [7,14].

The fundamentally different case examples show how the practical outcomes of projects dealing with large-scope issues are connected—in addition to the societal conditions— to the local situations. As in any case study, the case-specific prerequisites and constraints of change, if subjected to further study, will enhance the understanding of the phenomena in question. While Usher and Bryant (1989) [17] emphasise the situatedness of any enquiry, we can conclude that the way AR is carried will contribute to the situatedness and alter the potential for actual change to take place. This points to the need for continuous flexibility in planning and conducting the AR processes.

### 3.3. Learning by Doing: The Practitioners and Researchers in Collaboration

Along the course of the AR projects, the participants and researchers met numerous times and shared learning opportunities. However, even in the most participatory forms of AR, their roles are different, and this affects the learning needs and outcomes. Since the entire article is essentially about researchers learning about AR in large-scope issues, we want to focus here on some concrete aspects of learning through the cases.

As in all AR projects, it was noted in Case 1 that framing the agenda is of vital importance [44]—that is, circumscribing the phenomena to which attention is paid [45] (pp. 309–10) and on which the joint work is focused. From the start, the programme provided the principles for working and stressed self-directed learning, for which both the theoretical offerings and discussions would produce material. More specific framing took place mainly in the multiprofessional work groups. The basic idea was that the discussions would generate alternative ways of framing reality. At its best, it also functioned in this way and helped the participants to look at the situations with fresh eyes when the demands for change in the public sector started to intensify [46].

"We did have a joint thread in our small group, even though it was not visible all the time. In any case, we did have a similar frame."

"Receiving new perspectives—either through lectures or discussions—has been inspiring. When the topic was somehow familiar, speaking the same language but giving

something new—that is, not repeating how I myself think about it, but generating a problem, a new perspective—this was inspiring."

(Excerpts of two respondents in the interviews after the programme)

However, the path was not an easy one considering the different perspectives present in the discussions, and occasional drifting of the discussions could not be avoided. The research group took a conscious risk by leaving the structure, content, and methods partially open at the start. Such a structure is vulnerable to criticism if the participants have expectations based on more traditional educational programmes. Some participants also expressed their confusion, especially in the first half of the programme: "I wonder what will come out of this, but let's see." In retrospect, we think that, though risky, this was an appropriate strategy for the context and, perhaps also more generally, for large-scope issues that became even more significant during the deepening recession in Finland. In these circumstances, many of the issues to be resolved within the programme appeared in a new light and the discussions around them had to be reframed. Accommodation of the programme for the needs of the new situations required the research group to be alert both during the seminars and between them, and to bring to the forum research perspectives that might help to deal with the increased demands on the public sector in terms of policies and public discussion.

In the interviews after the programme, the participants pointed to the cognitive learning results in addition to the immediate effects on their work and the organisation. Some sort of support for agency was also valued in the midst of the public sector turmoil as is visible in these interview excerpts.

"It has opened up a new perspective and opportunities to examine issues of management."

"This [programme] has opened up horizons on this situation [of the public sector]."

"At first, I thought the programme should increase my resources in this organisational arrangement. I think it has done something like that."

Even though all the participants' focus was somehow on their work and the challenges of change, there was a clear difference in perspectives, which is characterised by Habermas's (1984; 1987) [22,23] differentiation of the lifeworld and the system. The system was the focus for participants who worked in administration, especially in generalist tasks, while for those in customer service whose focus was outside the system, the system requirements regulated the work. The programme was long enough for the participants to become familiar with each other, at least in the small groups. This made it possible for the lifeworld to be present in the discussions, in addition to the collisions between the system and the lifeworld, which in some cases were quite tense. The following extract is from an interview with a person who worked in personnel administration and whose participation in the programme took place in less than favourable conditions, partly because her immediate superior opposed her taking part in the programme:

"I expect it was me who was most anxious to bring up worries and troubles at work to the group . . . I did not find any very great wisdom to help my working community, but I did find things that helped me in my own position."

Overall, the atmosphere of the programme was safe enough to allow a multitude of perspectives to come up, as well as criticism of the organisers. In the groups, the mode of working developed towards a process that could be called reflective learning through challenging the perspectives of the other participants [34]. For example, discussions between ground-level professionals in customer service and general developers in central administration helped both sides to form a more realistic and practical picture of the situation and their context.

In organising the timetables, meeting rooms, and welcome coffees for the many communicative spaces, the supervisors and managers participating in Case 2 learned to understand that actual changes cannot be expected to occur without time and resource allocations. Time is a key resource: clients must also be attended to when the staff is involved in development projects; if time is not allocated, the staff will not participate.

In Case 2, resources were also allocated for training in computer skills and allergy issues. In addition, there is evidence that the kitchen staff and their supervisors learned to incorporate the learning opportunities and effect change in their work. One potential explanation for this may be the special context: the project was offered to them first in the day-care organisation. They were in a position to note that project gains only emerge when people start to involve themselves in interaction, communication, and learning, that is, to use their opportunities for empowerment and emancipation.

In Case 2, the researchers really "learned by doing," which is one of the basic challenges of AR—the significance of the initial negotiation phase of the project. It turned out that the chief shop steward, as the initiator of the kitchen project, had given all necessary information about the conditions of the kitchen staff and the management styles in the city, whereas the real gatekeepers in many other cases remained unknown for a long period of time. Even though the initiators of the projects would be top managers, the projects did not always gain legitimacy [42]. Wicks and Reason (2009) [25] have discussed the importance of these first steps in AR, and we have included the negotiation phase among the evaluation indicators of AR processes [37]. Acknowledging the role of professionalism also became necessary. In the cooperation between kitchen and other staff of the day-care centres, the core issue was the equality of all professional groups.

## 4. Discussion and Conclusions

The dual agenda of AR [30] calls for discussion and conclusions about the actions taken and the theories used. Furthermore, the fruitfulness of a parallel analysis of the two AR research lines of the working life research institute—sharing communicative theory as their vantage point and applying organisational learning approach—needs to be considered.

The descriptive analysis of the two AR lines adopted in the cases shows the versatile applicability of dialogue-based communicative spaces in dealing with large-scope issues in their societal contexts. In retrospect, we see that, although Case 1 was conducted during the 'moments' of AR [4] and that Case 2 relied more on the traditional idea of the evaluation of action taken after dialogues as a means of learning and making new action plans [47], the end result is the same: the participants learn when they reflect together on the initial opportunities to take new action and the outcome of the action taken. This was possible via the reflective practices and learning via the communicative spaces offered to them by the AR interventions. Case 1 offered the voluntary participants—adult learners—a further education programme that was conducted as a dynamic adjustment to emerging needs, whereas Case 2 was planned by negotiation and carried out change by deliberating different perspectives. Although restricted by these local situations, the participants in both AR lines had the opportunity to contribute to a process where communicative spaces, as a method of AR, turned out to also be means to learn to make changes together at the workplace [7].

However, the cases differ in terms of how the common agreement, in the spirit of Habermas [25,26], and the element of power [14]) are realised as the basis for concrete action. The differences culminate in the participants' challenge to learn to use their agency [38]. In Case 1, agency was needed to confront the organisational power in applying the learning in the participants' own organisations later, whereas in Case 2 the voice of all participants had the possibility to be heard immediately in the framework of the redistribution of organisational power on dialogue forums. These differences mean that the participants of Case 1 had to rely on their own emancipation and empowerment, whereas the participants of Case 2 often got their reward immediately in the form of joint action.

In our interpretation of the differences of the cases, we rely on the notion of Habermas (1984; 1987) [22,23] about lifeworld and the system. When an AR project deals with any current large-scope issue—for example, with ways to cope with retrenchments and fundamental changes in the modes of operation of professional organisations—the system seems to take over. However, all participants in the dialogues interpret this system from the point of view of their lifeworlds—namely, who they are, what they do, what kind of work-

related identities they possess [6]), and how they use their agency [38]. It can be concluded that, due to hierarchical positions, there are individual challenges in AR processes, which vary from learning to use one's agency to learning to hear other participants. However, it is learning that enables us to make changes together.

Our results highlight the observation that a communicative space must be a safe space so that participants do not hide their lifeworlds. They need to know that their pursuit of emancipation will not end with domination by the system. This need for safety is in agreement with the notions of both Martin (2008) [12] and Mead (2008) [40], who have conducted AR handling large-scope issues and confronted the challenges for researchers. In our example, the adult education programme provided an urgently needed forum for collective reflection, even at a very personal level.

Among the kitchen workers, the application of democratic dialogue, which emphasised the significance of everyone's work experiences and right to learn, brought the system and the lifeworld closer together. In professional and expert work, the worker in many cases carries out tasks alone without partners for reflection. It seems that, in such cases, there is a risk of a widening gulf between the lifeworld and the system in situations of drastic change. Based on the findings, the use of communicative spaces is justified in bringing together the system and the lifeworld, when proper care is taken about threats to free communication. This should not be impossible, since in adult education—as well as in AR in general—research may emerge only stepwise, via an empirical analysis of conditions and situations. In the context of AR, this cannot be done in isolation, solely by researchers. Instead, they need to be prepared for many setbacks and ready to modify their research plan if cooperation with the participants continues despite these setbacks.

We can conclude that our reflection contributes to the practice of AR by showing the malleability of the dialogue-based AR method. The method was initiated to develop workplace practices and promote organisational learning by learning together in dialogue conferences [9] and is also applicable in other frameworks to be linked to modifications of the AR cycle. From the point of view of the theory of AR, we have shown that the method has vital roots in the Habermasian [22,23] ideal of free communication and that the element of power, usually connected to domination, may be used to promote emancipation.

After reflecting on these AR cases dating back to the 1990s, we can see many connections to the present situation with its changing public sector and implications for organisational learning and adult education in the broad context of lifelong learning [48]. At the same time, we recognize a major limitation of our study: we have not discussed in detail the constraints and preconditions that different societal contexts set on communicative spaces. During our cases the Finnish, and already earlier the Norwegian and Swedish, societal contexts were favourable for this type of initiatives. Instead, we put our emphasis on the detailed characteristics of the communicative spaces and the versatile applicability of the method. However, also in the future, which is necessarily characterised by significant changes in working life as well as in society in general, we will need to deal with many large-scope issues in communicative spaces. These spaces enable participants to make the necessary changes in their work or to better adapt to the continuous changes of a globalising world, learning to be critical in both senses of the word [4,28]: being analytical and addressing the issue of power relations embedded in the matters to be developed.

The management of the large-scope issues inherent in the future challenges of sustainability would gain from dialogues conducted continuously in communicative spaces. This type of communicative space may be seen as a sustainable, permanent, although malleable, structure of organizational learning and development. It can involve people in organisational and societal changes through their experience and through learning from others. The spaces need to be supported by favourable social conditions, national and international. Individuals learn, nations learn, and cultures change when there is a right moment and active advocates of change. We have shown that there is more than one way to build communicative spaces, and that they are not topic specific. Being optimistic, it is

not impossible to learn to share the power embedded in local, national, and international societies, in a pursuit of a common goal.

**Author Contributions:** Conceptualization, S.K. and T.H.; methodology, S.K. and T.H.; investigation, S.K. and T.H.; writing—original draft preparation, S.K. and T.H.; writing—review and editing, S.K. and T.H. Both authors have read and agreed to the published version of the manuscript.

**Funding:** This research received no external funding.

**Informed Consent Statement:** Informed consent was obtained from all subjects involved in the first stages of this specific study that concentrates on publicly available materials.

**Data Availability Statement:** The data are listed in the references.

**Conflicts of Interest:** The authors declare no conflict of interest.

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
