# Peer review of "Experiences of Opening Up Communicative Spaces for Large-Scope Issues"

_challenges, doi:10.3390/challe12020025_

Round 1
Reviewer 1 Report
Review Report
Experiences of opening up communicative spaces for large-scope issues
Review
The paper describes and analyses two action research project conducted in Finland in the 1990s. With some major changes the paper will be ready to be published.
Major issues
The introduction would greatly benefit from subheadings. For example: Organizational Learning, Habermas and Critical Theory etc. Another suggestion is to present the aim of the study earlier on in the introduction and then go ahead to give the theoretical background.
Please make some additions and elaborations in the methods section:
- Describe the method of analysis used in the paper in the methods section. How were data analysed? Which steps were taken? You only give a very brief statement “this is a result of a qualitative second reading”. Please elaborate!
- For case 2 participants and time frame is presented (line 267-8). Please give the same information for case 1.
The results section should only contain results obtained from your analysis. Please refrain from discussing the results. See for instance line 390-404, line 512-514, and line 543-551.
Author Response
We thank the peer reviewers for their valuable comments and are happy to inform that we were able to take into account almost all of them. The revisions are highlighted as yellow in the resubmitted text.
- Review: With some major changes the paper will be ready to be published.
Major issues
The introduction would greatly benefit from subheadings. For example: Organizational Learning, Habermas and Critical Theory etc. Another suggestion is to present the aim of the study earlier on in the introduction and then go ahead to give the theoretical background.
The authors: Of these two options, we chose to present the (general) aim of the article in the lines 64-68.
Please make some additions and elaborations in the methods section:
- Describe the method of analysis used in the paper in the methods section. How were data analysed? Which steps were taken? You only give a very brief statement “this is a result of a qualitative second reading”. Please elaborate!
- The authors: Details of the analysis are given on the lines 258-271.
- For case 2 participants and time frame is presented (line 267-8). Please give the same information for case 1.
- The authors: The time line of Case 1 is added to the lines 279-282 and also the number of participants is on the line 282.
The results section should only contain results obtained from your analysis. Please refrain from discussing the results. See for instance line 390-404, line 512-514, and line 543-551.
The authors: We prefer not to make the suggested revisions of this point for the following reasons: Our article is composed more along qualitative and narrative traditions of social science than according to a strict IMRD-formula. Often the social sciences conventions and practices demand that the results are linked to the chosen theories and concepts already in the connection of their first presentation whereas discussion and conclusions widen the perspective towards more versatile interpretations and sometimes towards practical applications.
Reviewer 2 Report
This manuscript is a fine piece of work overall. When referring to macro level societal developments in the 1990s, you provide as context the new public management era and refer to staff downsizing, managerialism, marketization and privatisation. It should merit attention also, that the recession of the 1990s was especially severe in Finland (also in Sweden). The public sector staff was temporarily cut, but as total tax ratio increased, so the public sector increased in fact. Public sector staff cuts were recovered by the end of the 1990s. Public sector management principles are relevant on organisational level, but they were not the rationale for public sector temporary staff cuts in the 1990s' in Finland. You refer to research ethics very well (row 233). Here, it could be worth specifying what was the role of researchers in terms of independence in relation to the cases studied, especially participant observation method raises this issue. The cases are tied to theory in a really competent manner and the use and interpretation of literature is done in a mature way. In the references 10, 33, 34 and 37, the year of publication is missing.
Author Response
We thank the peer reviewers for their valuable comments and are happy to inform that we were able to take into account almost all of them. The revisions are highlighted as yellow in the resubmitted text.
2. Comments and Suggestions for Authors
This manuscript is a fine piece of work overall. When referring to macro level societal developments in the 1990s, you provide as context the new public management era and refer to staff downsizing, managerialism, marketization and privatisation. It should merit attention also, that the recession of the 1990s was especially severe in Finland (also in Sweden).
The authors: This fact (about Finland only), is added on the lines 117-118.
The public sector staff was temporarily cut, but as total tax ratio increased, so the public sector increased in fact. Public sector staff cuts were recovered by the end of the 1990s. Public sector management principles are relevant on organisational level, but they were not the rationale for public sector temporary staff cuts in the 1990s' in Finland.
The authors: We have added a notion of the wide discussions about the rationale of NPM matter among professionals on the lines 76-78.
You refer to research ethics very well (row 233). Here, it could be worth specifying what was the role of researchers in terms of independence in relation to the cases studied, especially participant observation method raises this issue.
The authors: We have added an explanation how the range of matters under observation was not limited by the participants and about mutual trust as a prerequisite for this on the lines 244-249 and give also some case specific details on the lines 367-368 and 376-377.
The cases are tied to theory in a really competent manner and the use and interpretation of literature is done in a mature way. In the references 10, 33, 34 and 37, the year of publication is missing.
The authors: We have added the missing publication years to the list of references.